# Language Modeling With Factorization Memory

## Abstract

We propose Factorization Memory, an efficient recurrent neural network (RNN) architecture that achieves performance comparable to Transformer models on short-context language modeling tasks while also demonstrating superior generalization in long-context scenarios. Our model builds upon Mamba-2, enabling Factorization Memory to exploit parallel computations during training while preserving constant computational and memory complexity during inference. To further optimize model efficiency and representational capacity, we develop a sparse formulation of Factorization Memory that updates only a subset of recurrent states at each step while preserving the strong performance of its dense counterpart. To our knowledge, this represents the first RNN architecture that successfully combines sparse memory activation with competitive performance across both short and long-context settings. This work provides a systematic empirical analysis of Factorization Memory in comparison to Transformer and Mamba-2 architectures.

## 1 Introduction

Transformer-based language modeling (Brown et al., 2020) has significantly advanced natural language processing (NLP) through multitask fine-tuning (Taori et al., 2023; Sanh et al., 2021). This paradigm shift has redefined NLP development, moving from training task-specific models to building general models capable of solving multiple tasks.

A particularly challenging frontier is ultra-long-context understanding, where traditional models encounter fundamental limitations. Long-context comprehension is essential for complex reasoning Yang et al. (2025b), software development Zheng et al. (2023), and multi-session conversations Maharana et al. (2024). The quadratic complexity of transformers, $O(n^2)$, remains a well-known bottleneck, posing significant challenges for scalability. Addressing this computational complexity requires forgoing the attention mechanism, which is the primary source of inefficiency.

Recently, there has been increasing interest in revisiting the recurrent neural networks (RNNs) due to their bounded memory requirements and linear generation complexity. State-space models (SSMs) (Gu & Dao, 2024) have inspired parallelization-enabled RNNs, making them viable competitors to Transformer. This led to the development of modern recurrent architectures such as Mamba Gu & Dao (2024), Mamba-2 (Dao & Gu, 2024), MiniLSTM (Feng et al., 2024), Gated Linear Attention (Yang et al., 2024), RWKV Peng et al. (2023), and others. Unlike Transformers, rather than accessing the entire input sequence at inference, RNNs encode sequences into fixed-size recurrent states.

This compressive nature limits RNN performance on tasks requiring precise recall of long token sequences. RNNs encode an unbounded sequence into a fixed-size hidden state Oren et al. (2024), creating a bottleneck: finite bits must represent unlimited information. Precise recall over long spans (e.g. verbatim repetition of a random passage) is fundamentally difficult. Even though some models use time-dependent parameters and does very well on many memory tasks, this core limitation still applies when lossless recall is needed. Increasing the hidden state size can help, but at the cost of inference efficiency, potentially erasing RNNs' performance advantage over Transformers.

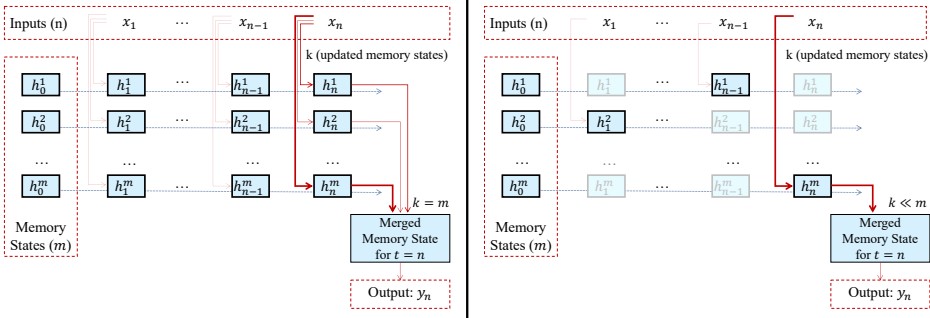

Figure 1: Factorization Memory - Layer Schematics. **Left:** In the *dense* formulation all $m$ memory states are updated at each timestamp. The updates are weighted with memory-input affinity scores, and the thickness of the arrows represents the strength of the update. **Right:** In the *sparse* formulation, only selected top-$k$ states are updated at each timestamp. Grey shading indicates that the state is used neither in update nor in merge operations.

To balance efficiency and capacity, we propose Factorization Memory, a novel RNN architecture that reconciles these competing objectives. Unlike Mamba or similar models, Factorization Memory employs sparse recurrent state updates, enabling selective updates of only a small subset of parameters at each time step (see Figure 1). This reduces the computational overhead associated with recurrent state updates, allowing larger recurrent states while maintaining a bounded computational cost. Unlike prior work that explores sparsity as a theoretical property Cheng et al. (2024); Liu et al. (2021), our method achieves compute and memory savings during training and inference thanks to partial activation.

Our empirical evaluations (see Section 4) show that not only Factorization Memory is competitive with Transformer and Mamba-2 on short-context tasks, but it also exhibits superior performance extrapolation beyond the training context length. Furthermore, it achieves higher inference efficiency than these models (see Section 4.2.3).

The contributions of this paper are as follows:

1. We introduce Factorization Memory, a recurrent memory model that demonstrates competitive performance on short-context tasks while outperforming Transformer and Mamba-2 in long-context extrapolation.

2. We propose sparse RNN memory mechanism that selectively updates only a subset of states, bounding the computation and memory cost while scaling up the model and maintaining a strong benchmark over its non-sparse counterpart.

3. We release optimized CUDA/Triton kernels for Factorization Memory, ensuring reproducibility and facilitating future research into sparse RNNs[1].

## 2 BACKGROUND

In this section, we will briefly review the fundamentals of Transformer and RNN architectures and their relationship with Factorization Memory.

**Transformer** A standard Transformer (Vaswani et al., 2017) layer can be expressed as a composition of self-attention, residual connections, and a feedforward MLP. Here, we focus on the self-attention mechanism, which models dependencies between input and output tokens. Given inputs $X$ projected onto queries, keys, and values ($Q$, $K$, $V$), attention in its simplest form can be expressed as follows[2]:

$$\text{Attention}(Q, K, V)_t = \frac{\sum_{i=1}^{N} e^{q_t^T k_i} \cdot v_i}{\sum_{i=1}^{N} e^{q_t^T k_i}},$$ (1)

The attention output is a linear combination of value projections up to time step $t$. This formulation can be interpreted as an RNN with an unbounded number of key-value states

---

[1]To comply with the double-blind review process, we will release the code upon acceptance.
[2]We use the latest attention architecture for our experiment baseline.

to which $q_t$ attends (Oren et al., 2024). In auto-regressive settings, self-attention represents the most expressive form of a multi-state RNN with infinitely growing state size.

**Recurrent Neural Networks** , Recurrent models such as Mamba-2 (Dao & Gu, 2024) maintain a fixed-sized recurrent state to represent the input sequence. These models follow the recurrence:

$$h_t = A_t h_{t-1} + B_t x_t \tag{2}$$

where $h_t$ is the recurrent state and $x_t$ is the input at time $t$. The forms of $A_t$ and $B_t$ determine model expressiveness and computational properties. In structured state-space models (SSMs) like Mamba-2, $A_t$ and $B_t$ do not depend on the recurrent state $h_t$ itself (though they may depend on previous layer hidden state). This recurrence can be parallelized using the **parallel prefix scan** algorithm (Blelloch, 1990), allowing scalable training while retaining the inference efficiency.

Gated Linear Attention (GLA) (Yang et al., 2024) introduces a gating mechanism to selectively control information flow in sequence models. Instead of treating all tokens equally, GLA applies multiplicative gating to dynamically regulate which information passes through. This gating mechanism allows finer control over long-range dependencies while maintaining efficient computation.

Inspired by Mamba-2's structured recurrence and Gated Linear Attention's selective gating, Factorization Memory adopts a similar approach to recurrence with a focus on computational efficiency and enhanced capacity. Factorization Memory constrains $A_t$ and $B_t$, ensuring that only a small portion of $h_t$ is updated and used at each time step.

## 3 Factorization Memory

The main intuition of Factorization Memory architecture is that a model should selectively choose and manage parts of hidden recurrent state. As shown in Figure 1, this architecture maintains a 2-dimensional recurrent state with $m$ rows (memory states). Upon receiving input $X = \{x_t\}_{t=1..n}$, memory states can be updated with two strategies: (1) *dense*: weighted update over memory states, and (2) *sparse*: selecting only $k$ memory states for update.

### 3.1 Dense Memory Update

At each step $t$, the dense Factorization Memory updates all $m$ memory states. Input $\bar{x}_t$ is put into memory proportionally to the affinity scores, defined as $\alpha_t = \text{softmax}(W_\alpha x_t)$. Formally, Factorization Memory recurrence can be expressed as follows:

$$h_t = \text{diag}\left(1 - \eta_t \alpha_t\right) h_{t-1} + \eta_t \alpha_t \otimes \bar{x}_t, \tag{3}$$

where $\eta_t = \sigma(w_\eta^T x_t)$ is the update rate governing the trade-off between state update and retention. When $\eta_t = 0$, the input is completely prevented from influencing the memory.

$\alpha_t$ essentially represents a conditional probability distribution over the memory states, quantifying their relevance to the input vector. A uniform $\alpha_t$ distributes the input evenly across all memory states, while a highly skewed $\alpha_t$ *factorizes* the input into a few selected states. From a capacity perspective, it is desirable for each memory state to encode information corresponding to distinct aspects of the input, essentially clustering the tokens by the "topics"; therefore, skewed updates are generally preferred. To control the sharpness of this distribution, we introduce a temperature parameter $\tau$ in our experiments, redefining $\alpha_t$ as softmax$(W_\alpha x_t/\tau)$ where appropriate.

The output of the layer is obtained by projecting the aggregated memory states. To merge $m$ separate memory states, we apply root mean square (RMS) normalization to each row of the recurrent state $h_t$, followed by computing a linear combination of the normalized states. More formally, $y_t = W_o \text{norm}(h_t)^T \phi_t$, where $W_o$ is the output projection matrix, and $\phi_t$ is the combination weights. To compute $\phi_t$, we reuse the affinity scores $\alpha_t$, ensuring that the same memory states involved in the update are also used in memory aggregation. This choice of $\phi_t$ is important for the Sparse Factorization Memory (see Section 3.2). Similarly to the

update rate, we introduce merge rate $\mu_t = \sigma(w_\mu^T x_t)$ to control the scale of the aggregation: $\phi_t = \mu_t \alpha_t$.

Combining all these components, the dense update in Factorization Memory is as follows:

$$\alpha_t = \text{softmax}(W_\alpha x_t) \in \mathbb{R}^m \qquad \text{memory affinity scores} \qquad (4)$$

$$\eta_t = \sigma(w_\eta^T x_t) \in (0,1) \qquad \text{shared update rate} \qquad (5)$$

$$\mu_t = \sigma(w_\mu^T x_t) \in (0,1) \qquad \text{shared merge rate} \qquad (6)$$

$$\theta_t = \eta_t \alpha_t \in \mathbb{R}^m \qquad \text{memory update weights} \qquad (7)$$

$$\phi_t = \mu_t \alpha_t \in \mathbb{R}^m \qquad \text{memory merge weights} \qquad (8)$$

$$\bar{x}_t = W_i x_t \qquad \text{input projection} \qquad (9)$$

$$h_t = \text{diag}\left(1 - \theta_t\right) h_{t-1} + \theta_t \otimes \bar{x}_t \qquad \text{memory update with linear recurrence} \qquad (10)$$

$$y_t = W_o \, \text{norm}(h_t)^T \phi_t \qquad \text{output projection of merged memory states} \qquad (11)$$

where $x_t \in \mathbb{R}^{d_{model}}$ is a token embedding, $w_\eta, w_\mu \in \mathbb{R}^{d_{model}}$ are trainable parameters for update and merge rates, $W_\alpha \in \mathbb{R}^{m \times d_{model}}$ is a trainable matrix for memory affinity, $h_t, h_{t-1} \in \mathbb{R}^{m \times d_{memory}}$ are the memory states, $W_o$ and $W_i$ are the projection matrices for adapting dimensions between $d_{model}$ and $d_{memory}$. $\sigma(\cdot)$ denotes the sigmoid activation.

Similar to Mamba-2, Factorization Memory can rely on the parallel prefix scan algorithm for efficient training, enabling scalability across long sequences. During inference, the layer achieves efficiency by maintaining only the most recent recurrent state, eliminating the need for recomputation over the full sequence (unlike Transformer).

### 3.2 Sparse Memory Update

While skewed dense memory updates *factorize* the input into a few selected states, computationally, all $m$ memory states are still updated, requiring $\mathcal{O}(md_{memory})$ operations (see Appendix A.2 for derivation). Unlike other RNN variants (Section 5), Factorization Memory uses the same probability distribution $\alpha_t$ for memory update (write) and output merge (read) (equation 4).

We take computational advantage by treating $\alpha_t$ as a router and selecting only top-$k$ most relevant memory states for both the memory read and write. Formally, we re-normalize $\alpha_t$ scores with the top-$k$ sparse mask as follows:

$$\gamma_t = \mathcal{T}(\alpha_t, k) \qquad \text{select top-}k \text{ relevant memory states, compute sparse 0-1 mask} \qquad (12)$$

$$\bar{\alpha}_t = \frac{\gamma_t \odot \alpha_t}{\gamma_t \alpha_t} \qquad \text{re-normalize affinity scores after applying top-}k \text{ mask.} \qquad (13)$$

$\bar{\alpha}_t$ contains the new affinity scores and the rest of Factorization Memory computations remain the same as in the dense version (see Section 3.1). The sparse mask reduce the compute operations by dropping all subsequent operations where $\gamma_t = 0$. Since we reuse the affinity scores in the update and merge operations, only $k$ memory states need to be loaded for every timestep $t$.

$k$ is a configurable parameter, which allows to balance computation with respect to the full capacity of $m$ memory states, with $k = m$ we recover the dense update in Factorization Memory, in the ideal case we want $k$ as small as possible to realize compute reduction. We explore the trade-offs of tuning $k$ in Section 4.1.3.

## 4 Experiments

### 4.1 Test Loss Evaluation

In the following experiments, we want to establish general properties of Factorization Memory model through the test loss evaluations.

### 4.1.1 Model Settings, Pre-Training and Evaluation Datasets

We adopt the modern Transformer architecture as in (Touvron et al., 2023) and simply replace the attention layers with their Factorization Memory counterparts. We also benchmark our model against Mamba-2[3], an exemplar of the modern RNN family. For Transformer model, we are using Flash Attention 2 Dao (2024) during training and testing.

**Train/Test Dataset** We pre-train and evaluate the language models on a curated sample of Web data predominantly comprising English and Japanese texts. The dataset is filtered to ensure high-quality content, with an approximate size of 250B tokens. Following established practices (Penedo et al., 2023; Li et al., 2024a), our filtering pipeline removes duplicates, excessively short or long sequences, and low-information content with classification-based filtering. To ensure the language balance we employ fastText language identification models (Joulin et al., 2016b;a). For the test loss evaluations we reserve a random subset of this dataset. For training, we compose smaller subsets for each model following compute-optimal training regime of $\geq 20$ tokens per parameter (Hoffmann et al., 2022).

**Long Context Dataset** To evaluate long-context capabilities, we construct a benchmark of 1,000 English and 1,000 Japanese documents, exceeding 128K tokens each. These texts are sourced from publicly available Web novels to preserve coherence and linguistic consistency over extended contexts. To prevent data contamination, we ensure the evaluation samples do not overlap with the training corpus by sourcing it with later cutoff dates that those used in training, in addition to applying exact and fuzzy deduplication.

### 4.1.2 Long Context Scaling Law

When designing a new model architecture, it is essential to assess whether performance remains predictable as the model scales. We conduct experiments across multiple model sizes, analyzing Factorization Memory test loss (see Figure 3a). We compare its scaling behavior against Transformer and Mamba-2 baselines under identical training conditions (see Figures 2b and 2c). Transformer uses grouped query attention mechanism based on LLaMA architecture Touvron et al. (2023) and trained and tested using Flash Attention 2 Dao (2024). Mamba-2 implementation is from the original repository [4]. All models are trained on the context size of 1024 tokens and evaluated on the same 1024-token window.

Figure 2 presents test loss curves across different model architectures, sizes, and hyperparameters. For each model, we identify the Pareto frontier of the test loss function — termed the *Loss Frontier* — plotted against forward pass FLOPS.[5] Figure 3a shows how this frontier evolves with scale.

Factorization Memory exhibits predictable performance improvements with scale, mirroring Transformer and Mamba-2 models. Its loss frontier is shifted upward compared to Transformer and Mamba-2, which suggests that, under the same training conditions, it requires slightly more compute to achieve comparable test loss on 1024-token window.

The central design objective of Factorization Memory is to efficiently process long contexts through increased capacity. We examine extrapolation to extended context by assessing performance on a 2048-token context window and compute the corresponding loss frontiers (see Figure 3b). We filter out test samples with fewer than 2048 tokens to avoid bias from short sequences. Factorization Memory demonstrates a scaling trend in the loss frontier, while Transformer and Mamba-2 exhibit a lower degree of extrapolation. Mamba-2 generally achieves better long-context generalization than Transformers, likely due to its recurrent architecture. However, Factorization Memory surpasses Mamba-2 as more training FLOPS are allocated to it, showcasing superior scaling in context extrapolation.

To validate our context extrapolation findings, we analyze the mean test loss across all context lengths up to 128K tokens computed on *Long Context Dataset* (see Figure 4). Trans-

---

[3]We use Mamba-2 as in: `https://huggingface.co/state-spaces/mamba2-1.3b`.

[4]https://github.com/state-spaces/mamba

[5]Although total amount of compute should include backward pass, for simplicity we report only forward pass FLOPS. The backward pass during training can be approximated of 4x of forward pass.

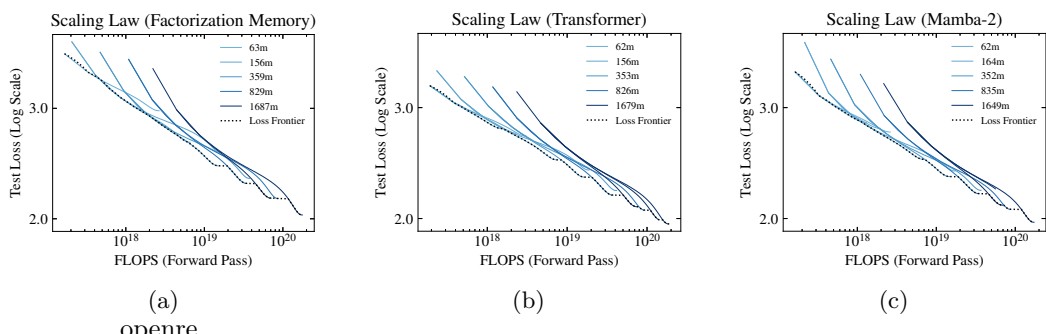

(a)  (b)  (c)

openre

Figure 2: Loss Frontier: All models are trained with the context length of 1024 tokens, while varying the number of model parameters, learning rate, and training budget.

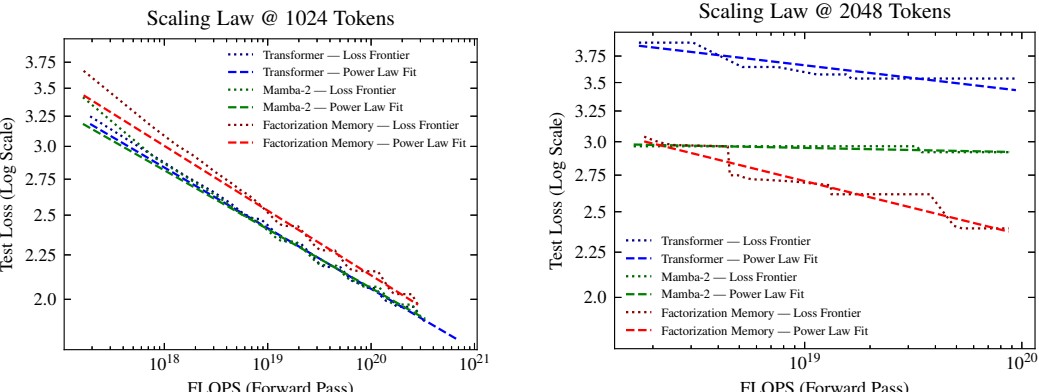

(a) The models are trained and evaluated on the context window of 1024 tokens. All the tested models consistently improve their test loss as more FLOPS are allocated to training.

(b) The models are trained on with context window of 1024 tokens but evaluated on 2048. Factorization Memory shows a consistent test loss improvement with increasing training FLOPS, outperforming Mamba-2 and Transformer.

Figure 3: Loss Frontier Scaling

former and Mamba-2 exhibit limited extrapolation to longer contexts, with test loss rising sharply beyond their 1024-token training window. While Factorization Memory also experiences increased test loss beyond 1024 tokens, this increase rarely exceeds the uncertainty in predicting the first 128 tokens.

### 4.1.3 MEMORY SCALING

We systematically investigate two key aspects of Factorization Memory design and their impact on performance. First, we examine whether increasing memory states $m$ improves performance. We hypothesize that a wider memory enhances capacity to store and retrieve information, leading to better representations and more effective learning. Second, we explore the benefits of a sparse memory formulation (see Section 3.2): a sparse representation could improve computational efficiency while maintaining performance.

Figure 5 presents results from 60–70 million parameters model. We systematically increase the number of memory states $m$ by powers of two and evaluate test loss performance. All models are trained under the same conditions and exposed to the same sequence of training instances. To enforce a skewed $\alpha_t$ allocation and bring about a sparsity component in memory updates, we experiment with the temperature $\tau$: $\alpha_t = \text{softmax}(W_\alpha x_t / \tau)$. For each $m$, we report results for the optimal $\tau \in \{1, 0.5, 0.25, 0.125\}$, optimized via grid-search.

For the dense memory, the results in Figure 5 indicate a clear trend of improvement in test loss as the number of states increases. Specifically, the model demonstrates a consistent reduction in test loss suggesting increased capacity.

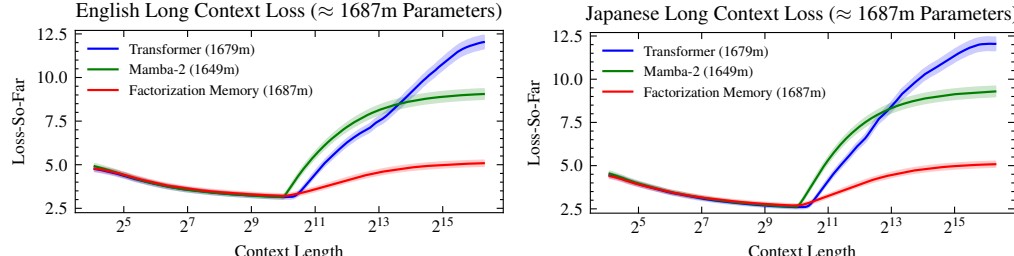

Figure 4: Loss-So-Far. **Left:** English. **Right:** Japanese. Factorization Memory consistently achieves performance comparable with Transformer and Mamba-2 at the training context length of $2^10 = 1024$ tokens. At the same time Factorization Memory shows better extrapolation at the long context. This pattern holds for all tested model sizes (see Appendix A.1).

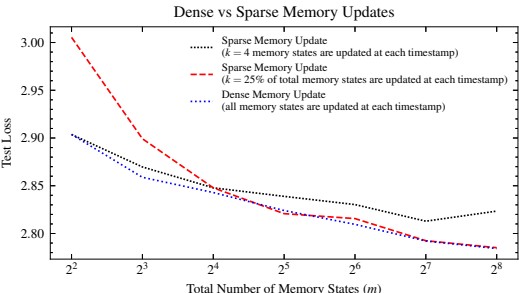

Figure 5: Comparison of *dense* and *sparse* memory updates: test loss generally decreases with an increasing number of memory states, even for sparse updates. Notably, updating only 25% of memory states achieves the same loss as dense formulation when the number of memory states is sufficiently large while reducing computational cost by 75%.

For the sparse memory, we investigate two distinct variants. In the first variant, referred to as *fixed memory activation*, updates are restricted to only 4 memory states per token, regardless of the number of states. In the second variant, termed *proportional memory activation*, only 25% of the total memory states are activated. While the fixed memory activation approach offers the best computational efficiency, it imposes strict constraints on updates, limiting the model performance. The proportional activation strategy still reduces the number of total operations and allows for greater flexibility in memory utilization.

Figure 5 reveals two key observations. First, increasing memory width improves performance for both sparse memory variants, albeit with diminishing returns in the fixed memory activation approach. The performance for fixed memory activation plateaus around a $m = 2^7 = 128$, whereas proportional memory activation continues to increase. According to the model definition, this shows that the router remained stationary across tokens. This outcome implies that each memory state encode distinct aspects of the input, achieving a non-redundant representation.

Second, proportional memory activation matches dense memory performance when the number of memory states is sufficiently large, suggesting that sparse memory activation is a more effective for scaling memory. Additionally, we observe that the optimal test loss requires a progressively lower temperature $\tau$ with increased states, supporting the benefits of skewed and sparse updates.

## 4.2 DOWNSTREAM TASK EVALUATION

In this section, we evaluate the efficacy of the model architectures on the standard English and Japanese LLM benchmarks (see Appendix A.4 for a detailed description of each). The benchmarks are composed of multiple-choice questions, which we evaluate in n-shot manner (except for IFEval (Zhou et al., 2023), where we follow the evaluation protocol specified in

| Model | Average | HellaSwag | MMLU | TQA | MUSR | IFEval | Winogrande |
|---|---|---|---|---|---|---|---|
| Transformer | 29.53 | **34.49** | 24.99 | 39.11 | 10.98 | 16.68 | 50.91 |
| Mamba-2 | 29.06 | 32.95 | **26.71** | 38.79 | 8.53 | 15.37 | 52.01 |
| **Factorization Memory** | **30.98** | 34.08 | 24.40 | **42.07** | **11.70** | **20.99** | **52.64** |

| Model | Average | JCS | JNLI | MARC-ja | xWino |
|---|---|---|---|---|---|
| Transformer | 56.41 | 48.70 | 41.95 | 80.99 | **54.01** |
| Mamba-2 | 49.73 | 37.53 | 34.43 | 75.01 | 51.93 |
| **Factorization Memory** | **59.80** | **53.80** | **47.49** | **84.08** | 53.81 |

Table 1: English (top) and Japanese (bottom) downstream evaluations: Factorization Memory achieves the highest average performance, outperforming Transformer and Mamba-2.

the original paper). The responses are selected based on the model's probability distribution, with the most probable answer (or answers) chosen given the prompt[6].

### 4.2.1 MODEL SETTINGS

To ensure fair comparison, we pre-train 1B-parameter models on identical data samples with the same training budget (Section 4.1.1 describes the pre-training dataset), attributing performance differences to architectures rather than training samples. We keep the models' hyperparameters the same where possible: all models for these experiments comprise 16 decoder layers, interleaving the target layer with MLP and residual connections. The model hidden size $d_{model} = 2048$. Factorization Memory uses $m = 64$ memory states with sparsity $k = 8$, and $d_{memory} = d_{model}$. The hyperparameters of Mamba-2 and self-attention layers are scaled accordingly to match the target model size (see also Section 4.1.1 for base settings).

### 4.2.2 RESULTS

The downstream evaluation results on English and Japanese are shown in Table 1. On the English tasks, Factorization Memory achieves the highest average score (30.98) performing best on TruthfulQA, MUSR, IFEval, and Winogrande. The Transformer model follows closely behind with an average score of 29.53, being slightly competitive on HellaSwag but underperforming on most of the other metrics. The Mamba-2 model, while performing better in MMLU, lags in overall performance (average score 29.06).

On Japanese metrics, the our proposed model once again leads with the highest average score (59.80), outperforming both Transformer (56.41) and Mamba-2 (49.73). It achieves the best performance on JCS, JNLI, and MARC-ja, indicating strong commonsense reasoning, natural language inference, and sentiment classification capabilities. Transformer performs best on xWino, while Mamba-2 consistently underperforms its competitors.

To ensure reproducibility of the experiments on the publicly available data, we also trained these 1B parameter models on the same 84B-token random sample of DCLM dataset Li et al. (2024b). The evaluation breakdown is available in Section A.3, Table 2. The experiments concur with our previous observation: on average, Factorization Memory performs competitively with both Transformers and Mamba-2.

### 4.2.3 INFERENCE SPEED

To evaluate the efficiency of 1B-parameters models, we benchmark inference speed on 16k token prompts, measuring the average generation time (see Figure 6). We use the optimized CUDA/Triton kernels for all the models, Key-Value (KV) cache on Transformer, and run experiments on a single H100 GPU (80GB). The results demonstrate that Factorization Memory consistently outperforms Transformer, whose quadratic complexity in sequence length leads to significant slowdowns, and it also exhibits a consistent 35-40% speed-up over Mamba-2, highlighting the efficiency of its sparse updates.

---

[6]We run Language Model Evaluation Harness framework for all evaluations https://github.com/EleutherAI/lm-evaluation-harness.

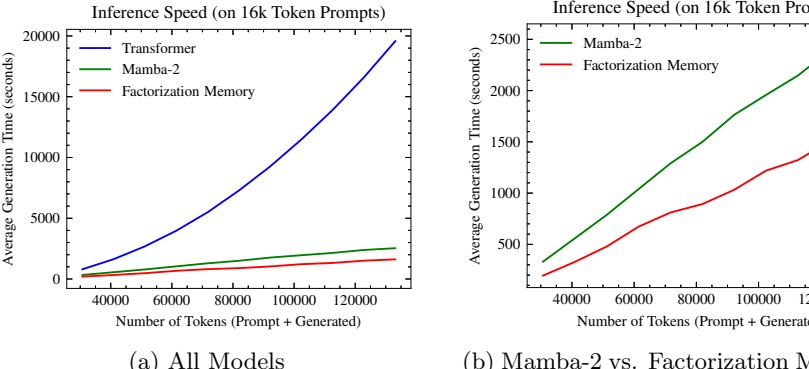

(a) All Models              (b) Mamba-2 vs. Factorization Memory

Figure 6: Inference Speed Comparison: Factorization Memory achieves better inference speed on long contexts than both Mamba-2 and Transformer models.

## 5 RELATED WORK

**Memory-Augmented Transformer**  Significant research has focused on augmenting Transformer with memory modules: Kang et al. (2025) for long context processing, Bulatov et al. (2022) for extended context retention, Ko et al. (2024) for temporal reasoning. Unlike these, our approach removes self-attention entirely.

**RNNs**  Factorization Memory is closely related to modern RNNs implementations such as RWKV (Peng et al., 2023), State-Space Models (Gu & Dao, 2024), Linear Attention Katharopoulos et al. (2020), and Gated Linear Attention (Yang et al., 2024; 2025a; Arora et al., 2025). In contrast to these models, our approach selectively updates small portions of the recurrent state, enhancing both efficiency and capacity.

**Hybrid Architectures** such as Hymba (Dong et al., 2024) and Griffin (De et al., 2024) combine the strengths of recurrence and attention model. We focus on "pure" architectures to isolate and evaluate model properties without confounding factors. Although a hybrid model is out the scope for this work, our approach can be a base for other architectures.

**Mixture of Experts** (Shazeer et al., 2017) introduces sparsity in MLP layers, whereas our method sparsifies along the time dimension, making the two approaches orthogonal.

**Accelerated Attention**  Several studies focus on accelerating attention through optimized implementations (Dao, 2023) or formulations (Liu et al., 2024; Zhang et al., 2022). While these approaches improve efficiency, they still incur quadratic complexity.

**Test-Time Training**  Recent test-time training approaches (Behrouz et al., 2024; Sun et al., 2024) adapt model parameters at inference, boosting capacity and mimicking memory. These approaches are orthogonal to Factorization Memory and compatible with our model.

**Transformer Adaptation to RNN**  Many recent approaches adapt attention layers into subquadratic analogs such as linear attention Zhang et al. (2025); Goldstein et al. (2025); Wang et al. (2025). While motivated by quadratic attention complexity, they focus on post-training adaptation of transformer models into existing linear recurrent architectures. Factorization Memory is compatible with these adaptation frameworks.

## 6 LIMITATION

The scope of this study is constrained by computational resources. This limits our investigation to relatively small-scale models with a low FLOPS budget. While our findings provide insights mostly concerning the test loss behavior, their generalization to larger models and more complex evaluations remains an open questions to address as future work.

## 7 CONCLUSION

We introduce Factorization Memory, an efficient RNN architecture that achieves performance comparable to Transformer and Mamba-2 models on short-context language modeling tasks while also demonstrating superior generalization in long-context scenarios. Factorized memory combined with sparse updates have proven effective in enhancing both the model efficiency and capacity in our experiments, offering a promising direction for research.

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

## A   APPENDIX

### A.1   LONG CONTEXT SCALING LAW

Figure 7 presents the complete set of long-context evaluations, extending Figure 4 (see Section 4.1.2 for experiment details).

### A.2   FLOPS CALCULATION

Let us denote $d_{model}$ as the hidden size of the model and $d_{memory}$ as the memory hidden size. The FLOPS introduced by each Factorization Memory layer per token can be approximated as follows.

$$
\begin{align}
& d_{memory}(2d_{model} - 1) && \text{input projection in equation 9} && (14) \\
& + m(2d_{model} - 1) && \text{memory affinity scores in equation 4} && (15) \\
& + 2(2d_{model} - 1) + 2m && \text{update and merge rates, } \theta_t \text{ and } \phi_t && (16) \\
& + m(4d_{memory} + 3) && \text{normalization for each memory state, norm}(h_t) && (17) \\
& + md_{memory} + d_{memory}(m - 1) && \text{memory merge in equation 10} && (18) \\
& + d_{model}(2d_{memory} - 1) && \text{output projection} && (19)
\end{align}
$$

If we exclude input and output projections and assume $d_{memory} = d_{model}$, then the recurrence update FLOPS can be bounded by $\mathcal{O}(md_{memory})$.

For the sparse formulation with top-$k$ selection, the compute reduction per-token can be estimated as $(m - k)(9d_{memory} + 5)$; the compute for the affinity scores remains unchanged.

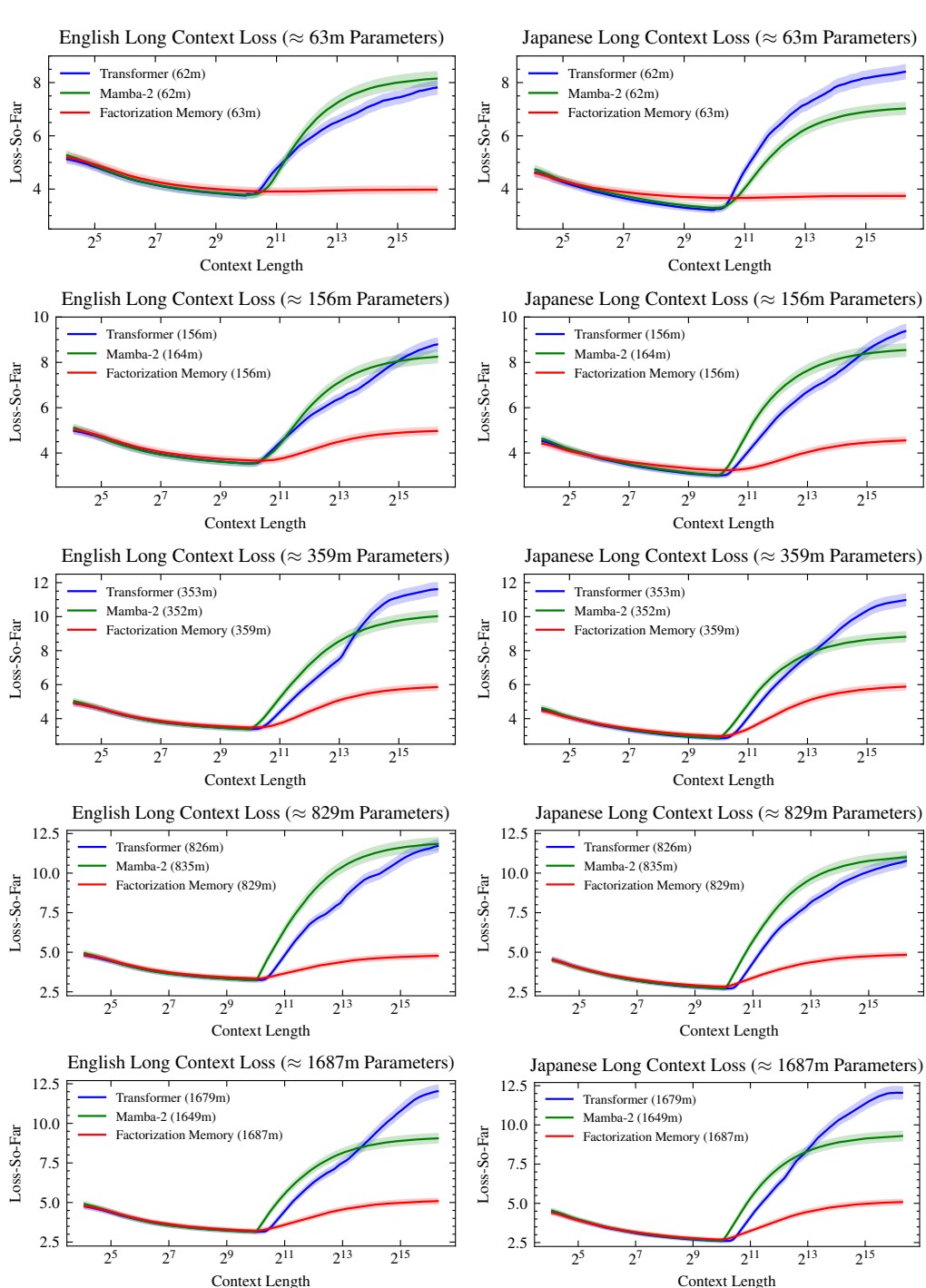

Figure 7: Loss-So-Far. **Left:** English evaluation. **Right:** Japanese evaluation. Factorization Memory model consistently achieves comparable performance at the trained context length of 1024 tokens with Transformer and Mamba-2 and at the same time shows better extrapolation at the long context.

| Model | Average | HellaSwag | MMLU | TQA | MUSR | IFEval | Winogrande |
|---|---|---|---|---|---|---|---|
| Transformer | 29.30 | 44.92 | 25.63 | 37.78 | 8.70 | 8.63 | 50.91 |
| Mamba-2 | 28.88 | 43.21 | **23.84** | 38.08 | 13.68 | 4.21 | 51.46 |
| **Factorization Memory** | **30.05** | **45.19** | 26.80 | **39.37** | **15.12** | **6.18** | **50.12** |

Table 2: English downstream evaluations on publicly available dataset, DCLM (DataComp for Language Model): Factorization Memory achieves the highest average performance, outperforming Transformer and Mamba-2. Results on Japanese are omitted, as DCLM is an English-language dataset.

## A.3 REPRODUCIBILITY NOTES

To ensure reproducibility of the experiments on the publicly available data, we trained 1B-parameter models on the same 84B-token random sample of DCLM dataset Li et al. (2024b). This subset corresponds to approximately 4x the compute-optimal budget, Hoffmann et al. (2022). We report the results on English downstream tasks on Table 2. Results on Japanese are omitted, as DCLM is an English-language dataset. On average, Factorization Memory performs competitively with both Transformers and Mamba-2, while maintaining superior inference speed as described on 4.2.3.

## A.4 EVALUATION BENCHMARKS

The English benchmarks used in our evaluation are as follows:

- **HellaSwag** Zellers et al. (2019) is a test set to benchmark model ability to perform commonsense reasoning, given questions that require understanding and reasoning beyond simple pattern matching.

- **MMLU** (Massive Multitask Language Understanding) Hendrycks et al. (2021) is a test to measure model multitask accuracy. It covers 57 tasks that covers history, computer science, mathematics, chemistry, and other topics.

- **TruthfulQA** Lin et al. (2022) measures models' inclination to replicate common online falsehoods

- **MUSR** (Multistep Soft Reasoning) Sprague et al. (2024) evaluates models on multistep reasoning tasks that simulate real-world decision-making scenarios. It measures the model's ability to reason over multiple pieces of information sequentially.

- The Instruction-Following Evaluation - **IFEval** Zhou et al. (2023) - is a benchmark designed to assess the proficiency of large language models (LLMs) in adhering to specific instructions.

- Winogrande Sakaguchi et al. (2019) is a large-scale dataset for commonsense reasoning, designed to be challenging by reducing linguistic biases and requiring deeper contextual understanding to resolve pronoun ambiguities.

The Japanese benchmarks included in our evaluation are as follows:

- **JCS** / JCommonSenseQA (Kurihara et al., 2022) is the Japanese version of CommonSenseQA (Talmor et al., 2019), which consists of multiple-choice questions that evaluate model ability in answering commonsense questions.

- **JNLI** / Japanese Natural Language Inference (Kurihara et al., 2022) is a test set to evaluate a model's ability to recognize the inference relation that a premise sentence has to a hypothesis sentence. There are three inference relations, namely: entailment, contradiction, and neutral which are presented to the model in a multiple-choice question format.

- **MARC** / Multilingual Amazon Review Corpus is a text classification test set that is based on the MARC dataset (Keung et al., 2020). MARC-ja is the Japanese subset of this dataset, which is a binary classification task for positive and negative review sentiment.

- **xWino**/ xWinograd-ja is the Japanese subset of the Winograd schema challenge (Emelin & Sennrich, 2021), which is a pair of sentences that differ in only one or two contrastive words that are easily disambiguated by a human reader. This evaluation measures model's ability to use commonsense knowledge to disambiguate and debias the ambiguous sentence.

