# OpenReview forum: "Language Modeling With Factorization Memory"
_ICLR.cc/2026/Conference — Submitted to ICLR 2026_

### Official Review · Reviewer_hwZ4 · 2025-10-26

**Soundness:** 2
**Presentation:** 2
**Contribution:** 2
**Rating:** 2
**Confidence:** 4

**Summary:**

This work aims to alleviate the limited retrieval and recall capability of Linear-RNN models caused by their fixed states. To this end, it introduces the concept of Factorization Memory, which selectively updates the memory slots within the state. The experiments analyze the impact of Factorization Memory on model loss and several common-sense reasoning tasks.

**Strengths:**

Factorization Memory is a sound idea that can effectively mitigate the memory conflict problem in Hebbian rule–based RNNs (such as Mamba2 and GLA).

**Weaknesses:**

W1: The idea of selectively updating memory slots within the state has already been widely adopted in many architectures, such as ABC [1] and GSA [2]. Moreover, the specific update formulation used in this paper is identical to that of models like HGRN2 [3] and MetaLA [4], yet no comparison with these models is provided.

W2: This paper attempts to alleviate memory conflicts by updating a subset of memory slots in the state using a top-k selection mechanism. However, models such as Mamba2 are already considered previous-generation Linear-RNNs, and many more advanced solutions now exist — including DeltaNet [5], Gated-DeltaNet [6], Comba [7], RWKV7 [8], TTT [9], Titans [10], and Lattice [11]. These models introduce delta-correction mechanisms to handle memory conflicts: physically, they apply an input-dependent cleanup to each row (each memory trace) of the state matrix S with respect to the current input k, preserving only the components orthogonal to k and thus retaining only new, important information. [7, 12] This approach is more efficient and expressive than top-k-based selection. Furthermore, techniques such as normalization, mentioned in this paper, have also been employed in the Lattice and TTT model families. Yet, the paper provides no discussion or comparison with these related works.

W3: For Linear-RNNs, inference efficiency is roughly similar across models; my main concern lies in pretraining efficiency. Prefix-scan parallelization is not hardware-efficient and cannot fully utilize modern tensor-core acceleration for matrix multiplications. The mainstream approach, as in DeltaNet, performs recursion across chunks rather than full-sequence scans. However, the introduction of normalization, top-k, and softmax operations in this paper makes chunk-level parallelization infeasible. The paper provides no analysis of pretraining efficiency or implementation details (e.g., Triton/CUDA kernels). I would suggest positioning Factorization Memory as a post-training technique instead, and exploring experiments on pretrained Linear-RNNs via distillation, which could be more valuable.

W4: The evaluation in this paper is insufficient. In particular, for small-scale models, the common-sense reasoning tasks shown in Table 1 have limited significance, since datasets smaller than 500B tokens do not expose the model to enough linguistic and factual diversity—most of what is learned is noise [13]. I recommend adding evaluations on classic synthetic benchmarks such as Ruler to provide a more meaningful assessment.


[1] Peng, Hao, et al. "ABC: Attention with bounded-memory control." arXiv preprint arXiv:2110.02488 (2021).

[2] Zhang, Yu, et al. "Gated slot attention for efficient linear-time sequence modeling." Advances in Neural Information Processing Systems 37 (2024): 116870-116898.

[3] Qin, Zhen, et al. "Hgrn2: Gated linear rnns with state expansion." arXiv preprint arXiv:2404.07904 (2024).

[4] Chou, Yuhong, et al. "MetaLA: Unified optimal linear approximation to softmax attention map." Advances in Neural Information Processing Systems 37 (2024): 71034-71067.

[5] Yang, Songlin, et al. "Parallelizing linear transformers with the delta rule over sequence length." Advances in neural information processing systems 37 (2024): 115491-115522.

[6] Yang, Songlin, Jan Kautz, and Ali Hatamizadeh. "Gated delta networks: Improving mamba2 with delta rule." arXiv preprint arXiv:2412.06464 (2024).

[7] Hu, Jiaxi, et al. "Comba: Improving Nonlinear RNNs with Closed-loop Control." arXiv preprint arXiv:2506.02475 (2025).

[8] Peng, Bo, et al. "Rwkv-7" goose" with expressive dynamic state evolution." arXiv preprint arXiv:2503.14456 (2025).

[9] Sun, Yu, et al. "Learning to (learn at test time)." arXiv preprint arXiv:2310.13807 (2023).

[10] Behrouz, Ali, Peilin Zhong, and Vahab Mirrokni. "Titans: Learning to memorize at test time." arXiv preprint arXiv:2501.00663 (2024).

[11] Karami, Mahdi, and Vahab Mirrokni. "Lattice: Learning to efficiently compress the memory." arXiv preprint arXiv:2504.05646 (2025).

[12] Siems, Julien, et al. "Deltaproduct: Improving state-tracking in linear rnns via householder products." arXiv preprint arXiv:2502.10297 (2025).

[13] Physics of Language Models: Part 4.1, Architecture Design and the Magic of Canon Layers

**Questions:**

If the authors are able to address the issues I raised above, I would be glad to engage in further discussion and consider raising my score.

---

### Official Review · Reviewer_XEe4 · 2025-10-29

**Soundness:** 2
**Presentation:** 2
**Contribution:** 2
**Rating:** 2
**Confidence:** 4

**Summary:**

This paper introduces an RNN architecture called Factorization Memory (FM). Its core innovation is that FM maintains multiple RNN hidden states (memories) and routes input tokens to the 'top-k' most relevant hidden states for updates. This approach employs sparse memory activation, and memories that are not selected remain unchanged. FM utilizes Mamba-2 as its memory update rule. The authors claim that FM performs well while being more efficient than dense activation. However, this concept has been introduced in former works.

**Strengths:**

1. This paper aims to solve the problem that linear models often struggle to compress long sequences into a fixed-size memory. Their performance on long-context tasks, such as NIAH and recall-intensive tasks, typically falls behind Transformers. This is a well-known and valuable problem.
2. FM shows better performance on long-context tasks, verifying the benefits of sparse memory activation and increased memory capacity.
3. FM can be more efficient than dense activation of memories.

**Weaknesses:**

1. The key point of this paper highly overlaps with MoM [1]. Both methods adopt sparse activation of memories with a router to direct different input tokens to the top-k memories. The theoretical framework was introduced in MoM, and FM seems like a special case that adopts Mamba-2 as the memory update rule.
2. The experiments are conducted using Mamba-2, but while FM could be applied to other linear models to verify its effect, this paper lacks such validation on other architectures (e.g., GLA, DeltaNet).
3. This paper claims that different memories are factorized, but the experiments are all focused on task performance. I think it would be better to include an analysis of how these memories are factorized and what the role of specific memories is.

[1] MoM: Linear Sequence Modeling with Mixture-of-Memories

**Questions:**

1. Could the authors add a discussion to the manuscript clarifying the relationship between FM and MoM, and detailing the specific novel contributions of this work?
2. The paper claims the model 'factorizes' inputs into topics. What empirical evidence supports this claim beyond simple sparse routing?
3. The speed comparison is conducted on sequences longer than 32k; what are the results on short-length sequences?

---

### Official Review · Reviewer_9EUs · 2025-11-01

**Soundness:** 2
**Presentation:** 2
**Contribution:** 1
**Rating:** 2
**Confidence:** 4

**Summary:**

This paper propose Factorization Memory, a RNN achitecture combining with sparse memory activation for improving model performance by selectively activating a specified number of memory states.

**Strengths:**

The motivation and design for extending model sparsity to the memory mechanism of linear model architecture is reasonable.

**Weaknesses:**

1. The authors claim that "To our knowledge, this represents the **first** RNN architecture that successfully combines sparse memory activation." However, similar sparse memory activation ideas have been explored in other works, such as MoM[1], SEE[2], etc. Further comparison and discussion with these works are needed.
2. The method builds on Mamba2, but lacks other more advanced RNNs as baselines, such as GLA[3], DeltaNet[4], and Gated DeltaNet[5]. Can proposed method be used in other RNN models with memory updating mechanism and be still effective?
3. According to Figure 2, the loss of the proposed Factorization Memory method does not seem to be better than Mamba2, and even the introduction of memory sparsity does not bring about an efficiency improvement.
4. The authors claim to have released optimized CUDA/Triton kernels, but I have not seen any related code implementation and am skeptical of its efficient implementation.

[1] Jusen Du, et al. MoM: Linear Sequence Modeling with Mixture-of-Memories.  arXiv, 2025.
[2] Yuqi Pan, et al. Scaling Linear Attention with Sparse State Expansion. arXiv, 2025.
[3] Songlin Yang, et al. Gated Linear Attention Transformers with Hardware-Efficient Training. ICML, 2024.
[4] Songlin Yang, et al. Parallelizing Linear Transformers with the Delta Rule over Sequence Length. NeurIPS, 2024.
[5] Songlin Yang, et al. Gated Delta Networks: Improving Mamba2 with Delta Rule. ICLR, 2025.

**Questions:**

See weaknesses.

---

### Official Review · Reviewer_fb5C · 2025-11-01

**Soundness:** 2
**Presentation:** 1
**Contribution:** 2
**Rating:** 2
**Confidence:** 5

**Summary:**

This paper introduces "Factorization Memory," a novel recurrent neural network (RNN) architecture designed to combine the strengths of Transformers and existing RNNs like Mamba-2. The core innovation lies in its ability to handle long-context language modeling tasks efficiently while maintaining competitive performance on short-context tasks. Factorization Memory achieves this by employing a 2-dimensional recurrent state with sparse updates, selectively activating only a subset of memory states at each time step. This sparse formulation aims to reduce computational and memory overhead during training and inference, allowing for larger recurrent states without sacrificing efficiency. The paper presents empirical analyses comparing Factorization Memory to Transformer and Mamba-2 architectures, highlighting its superior generalization in long-context scenarios and higher inference efficiency due to sparse updates.

**Strengths:**

1. Factorization Memory offers a genuinely novel approach to address the long-standing challenge of long-context understanding in RNNs. By introducing a 2D memory state with sparse updates, it attempts to reconcile the trade-off between memory capacity and computational efficiency.
2. The concept of sparse memory activation, where only a subset of recurrent states are updated at each step, is a significant strength. This directly tackles the computational and memory bottlenecks of dense recurrent states, especially as models scale. The paper explicitly states that this is "the first RNN architecture that successfully combines sparse memory activation with competitive performance."
3. The empirical results (e.g., Figure 4) show that Factorization Memory exhibits better extrapolation capabilities in long-context settings compared to both Transformer and Mamba-2, which is a critical advantage for handling very long sequences.
4. The paper demonstrates that Factorization Memory achieves higher inference efficiency, particularly on long contexts, outperforming both Transformer (due to its quadratic complexity) and Mamba-2 (due to its sparse updates). The release of optimized CUDA/Triton kernels further supports this claim and aids reproducibility.

**Weaknesses:**

1. "MoM: Linear Sequence Modeling with Mixture-of-Memories" is a good work also focus on sparsely expanding RNN memory states, but this paper did not compare with it. I recommend the authors to compare their algorithm difference and experimental performance.
2. The primary limitation stated by the authors themselves is the constraint of computational resources, leading to investigations on "relatively small-scale models with a low FLOPS budget" (60-70 million parameters for memory scaling, 1B parameters for main results). While the findings provide insights, the generalization to much larger models (e.g., tens or hundreds of billions of parameters) and more complex evaluations remains an open question.
3. While Figure 1 (Right) illustrates "only selected top-k states are updated" and Section 3.2 mentions k as a configurable parameter, there isn't an extensive ablation study or deeper analysis on the optimal choice of k or its sensitivity for various tasks and model sizes beyond the general statement that "in the ideal case we want k as small as possible." Figure 5 only focuses on m and τ.
4. While the mathematical formulations are provided, a clearer, more intuitive explanation or a block diagram detailing how the "sparse mask" is applied and how at acts as a "router" for both memory read and write could enhance understanding for readers less familiar with the specific mathematical notation.
5. While the paper mentions Mixture of Experts (MoE) as orthogonal and touches upon memory-augmented Transformers, a more explicit comparison of Factorization Memory's sparse update mechanism to other sparsity-inducing or memory-aware RNNs (e.g., those using external memory) would strengthen the discussion of its unique advantages.
6. The paper highlights inference efficiency as a strength. However, a more direct comparison of training time/costs between Factorization Memory, Transformer, and Mamba-2, beyond just FLOPS, would be beneficial. While sparse updates reduce FLOPs, the overhead of selecting top-k states could introduce other costs.
7. Figure 3a indicates that Factorization Memory requires "slightly more compute to achieve comparable test loss on 1024-token window" compared to Transformer and Mamba-2. While it surpasses them in extrapolation, this initial "cost" for comparable performance needs a more thorough explanation, especially in light of the overall efficiency claims.
8. The term "Factorization Memory" implies a particular structure or operation. While at being a conditional probability distribution that "factorizes the input into a few selected states" is mentioned, further elaboration on the "factorization" aspect and how it contributes to the model's capabilities would be useful.

**Questions:**

see Weaknesses

---

### Meta-Review · Area_Chair_mK6L · 2026-01-06

**Summary:**

Various concerns were raised in the initial reviews

- Lack of novelty
- Lack of discussion with relevant works
- Limited to small scale models
- Lack of ablation study
- Insufficient analysis
- Inferior empirical performance

**Reviewer Concerns:**

The authors did not provide a rebuttal.

**Reviewer Scores:**

Scores (2, 2, 2, 2) will not change due to lack of rebuttal.

---

### Decision · Program_Chairs · 2026-01-26

Reject